# Defining the Failure of Medical Therapy for Inflammatory Bowel Disease in the Era of Advanced Therapies: A Systematic Review

**DOI:** 10.3390/biomedicines11020544

**Published:** 2023-02-13

**Authors:** Monica State, Lucian Negreanu

**Affiliations:** 1Department 5, Internal Medicine—Gastroenterology, Carol Davila University of Medicine and Pharmacy, 020021 Bucharest, Romania; 2Colentina Clinical Hospital, 020125 Bucharest, Romania; 3Emergency University Hospital, 050098 Bucharest, Romania

**Keywords:** inflammatory bowel disease, Crohn’s disease, ulcerative colitis, failure, biologics, medical treatment

## Abstract

Background: The expansion of advanced therapies for inflammatory bowel disease created a lag between the development of these new therapies and their incorporation and use in daily practice. At present, no clear definitions for treatment optimization, treatment failure or criteria to abandon therapy are available. We aimed to centralize criteria for a nonresponse to all available molecules and to summarize guideline principles for treatment optimization. Methods: We conducted a systematic review of studies that reported criteria for the treatment response to all advanced therapies (infliximab, adalimumab, golimumab, ustekinumab, vedolizumab and tofacitinib) in patients with inflammatory bowel disease. Results: Across trials, criteria for a response of both patients with ulcerative colitis and Crohn’s disease are heterogenous. Investigators use different definitions for clinical and endoscopic remission, and endoscopic response and outcomes are assessed at variable time points. Current society guidelines provide heterogenous recommendations on treatment optimization. Most available data on loss of response concern anti-TNF molecules, and newer therapies are not included in the guidelines. Conclusion: The lack of clear definitions and formal recommendations provide the premise for empirical treatment strategies and premature abandonment of therapies.

## 1. Introduction

After the approval of tumor necrosis factor antagonists (anti-TNFs) for IBD treatment two decades ago, several other therapeutic options became available. With the development of advanced therapies, we are now witnessing a change in the management paradigm, as vedolizumab (VDZ), ustekinumab (USTEK) and tofacitinib (TOFA) are now providing new advances in achieving treatment goals for patients with IBD. However, a lag exists between the development of these new therapies and their incorporation and use in daily practice, as well as their incorporation in guidelines and recommendations.

A treatment nonresponse to biologics may be classified as primary or secondary based on the presence of an initial response. A primary nonresponse is generally considered if the drug was ineffective, with no clinical response within the initial treatment period, while a secondary nonresponse or loss of response (LOR) is considered if the effectiveness is lost over time after an initial response.

Criteria for assessing the response (definition and time frames) are heterogeneous and usually based on clinical and endoscopic activity scores. Clinical trial methodology includes clear definitions for responders, and a primary nonresponse is indirectly defined by subjects failing to meet the proposed outcomes. For the secondary loss of response, investigators propose variable clinical and endoscopic scores. For example, Roblin et al. [1] defined clinical failure for patients with CD as a Harvey–Bradshaw index ≥ 5 associated with a fecal calprotectin level > 250 µg/g of stool and for patients with UC as a Mayo score > 5 with endoscopic subscore > 1 or as the occurrence of adverse events requiring the patient to stop treatment.

Despite these general considerations, a consensus on how to optimize treatment, when to abandon a molecule and when to switch to another has not been reached. Current evidence is highly heterogeneous, and the lack of formal recommendations makes medical decisions difficult.

We aimed to discuss and centralize criteria for a nonresponse for all available molecules. Second, we summarized guideline principles for treatment optimization and recommendations on how and when to integrate new molecules into treatment strategies.

## 2. Materials and Methods

### Rationale

We conducted a systematic review of studies that assessed the treatment response of patients with IBD to all advanced therapies (infliximab, adalimumab, golimumab, ustekinumab, vedolizumab and tofacitinib).

Literature search. A structured search of the PubMed (MEDLINE) database was conducted on 1 December 2022. Our search terms included the following medical subject headings (MeSH) and text words:

For IBD: Inflammatory bowel disease, ulcerative colitis and Crohn’s disease

For treatment: anti-TNF, infliximab, adalimumab, golimumab, vedolizumab, ustekinumab, tofacitinib, anti-integrin, anti-interleukin and JAK inhibitors

First, we aimed to centralize definitions for a primary nonresponse. We extracted term definitions for clinical and endoscopic responses during the induction phase from all clinical trials and RCTs (randomized controlled trials). Subjects failing to achieve at least a clinical response after induction therapy were considered primary nonresponders.

The entire search algorithm was as follows: ((“inflammatory bowel diseases” [MeSH Terms] OR (“inflammatory” [All Fields] AND “bowel” [All Fields] AND “diseases” [All Fields]) OR “inflammatory bowel diseases” [All Fields] OR (“inflammatory” [All Fields] AND “bowel” [All Fields] AND “disease” [All Fields]) OR “inflammatory bowel disease” [All Fields] OR (“colitis, ulcerative” [MeSH Terms] OR (“colitis” [All Fields] AND “ulcerative” [All Fields]) OR “ulcerative colitis” [All Fields] OR (“ulcerative” [All Fields] AND “colitis” [All Fields])) OR (“crohn disease” [MeSH Terms] OR (“crohn” [All Fields] AND “disease” [All Fields]) OR “crohn disease” [All Fields] OR “crohn s disease” [All Fields])) AND (“anti-TNF” [All Fields] OR (“infliximab” [MeSH Terms] OR “infliximab” [All Fields] OR “infliximab s” [All Fields]) OR (“adalimumab” [MeSH Terms] OR “adalimumab” [All Fields]) OR (“golimumab” [Supplementary Concept] OR “golimumab” [All Fields] OR “golimumab s” [All Fields]) OR (“vedolizumab” [Supplementary Concept] OR “vedolizumab” [All Fields]) OR (“ustekinumab” [MeSH Terms] OR “ustekinumab” [All Fields]) OR (“tofacitinib” [Supplementary Concept] OR “tofacitinib” [All Fields] OR “tofacitinib s” [All Fields]) OR (“anti” [All Fields] AND (“integrin s” [All Fields] OR “integrins” [MeSH Terms] OR “integrins” [All Fields] OR “integrin” [All Fields])) OR (“janus kinase inhibitors” [Pharmacological Action] OR “janus kinase inhibitors” [MeSH Terms] OR (“janus” [All Fields] AND “kinase” [All Fields] AND “inhibitors” [All Fields]) OR “janus kinase inhibitors” [All Fields] OR (“jak” [All Fields] AND “inhibitor” [All Fields]) OR “jak inhibitor” [All Fields]) OR “anti-interleukin” [All Fields])) AND ((clinical trial [Filter] OR randomized controlled trial [Filter]) AND (fft [Filter]))

Inclusion criteria. All clinical trials and RCTs (randomized controlled trials) available as full text, either via open access or pay per view, were included in the screening process.

Exclusion criteria. Studies of pediatric populations, maintenance studies, post hoc analyses, experimental studies (e.g., local administration of IFX for fistulizing CD), lack of outcome definitions and studies not reporting on clinical/endoscopic outcomes were excluded.

Data extraction. Using a designed extraction form, two independent reviewers screened the filtered results and collected data on the type of study and main findings. The reference sections of the included studies were analyzed to retrieve relevant studies not identified in the original search.

Our review was registered on PROSPERO (ID 395550) and is currently awaiting publication on the registry site. Reporting of the review was conducted using the PRISMA guidelines.

## 3. Results

The search strategy for a primary nonresponse yielded 653 citations, of which 38 were included in the full-text analysis (Figure 1).

The characteristics of the trials are summarized in Table 1 (UC) and Table 2 (CD). Across trials, criteria for a clinical response in both patients with UC and CD are heterogenous.

All study protocols, except NCT00787202 [11], defined endoscopic remission/response as a Mayo endoscopic subscore ≤ 1. For patients with CD, endoscopic outcomes were not included in the protocol or were evaluated using various definitions.

All trials define clinical response by a decrease in the total Mayo score of at least ≥30%. Some trials use additional criteria, with an accompanying decrease of ≥1 point on the rectal bleeding component of the Mayo scale or a rectal bleeding subscore of 0 or 1 or they include the use of corticosteroids in the definition [7].

For Crohn’s disease, definitions are more heterogenous. Various clinical and endoscopic activity scores are used across trials. There is a general consensus regarding clinical remission, defined by a CDAI score <150. However, clinical response is defined by a wide range of changes in CDAI score, for different clinical settings. A similar situation is observed for the definition of endoscopic remission. Investigators proposed various activity scores (CDEIS, SES-CD) or used none and defined endoscopic remission only by absence of ulcerations [17]. Time frames for outcome assessment are variable, ranging from 4 to 30 weeks.

We further summarized current society guidelines and recommendations for treatment optimization strategies (Table 3).

## 4. Discussion

### 4.1. Therapeutic Response and Clinical Trial Endpoints

The treatment response is assessed through predefined endpoints using variable cutoffs for commonly used scores (Mayo, CDAI, SES-CD, and CDEIS). If the use of the Mayo score (both clinical and endoscopic subscores) is preferred to other criteria for assessing response to medication in the majority of clinical trials, definitions and scores vary widely for CD patients. Investigators use different definitions for clinical and endoscopic remission, and endoscopic response and outcomes are assessed at variable time points (Table 1 and Table 2). Previous studies reported that increasing the stringency of clinical and endoscopic endpoint definitions in CD trials, particularly lowering stool frequency or SES-CD definitions, reduces the ability to detect treatment-related changes in CD activity [27].

Before the emergence of new molecules for IBD treatment in real-life settings, the decision to stop biologic therapy was made based on solid evidence of a nonresponse, usually beyond the induction period. At present, considering therapeutic alternatives, clinicians may be prone to deciding on a premature switch of therapy. Clear and consistent recommendations are needed to guide treatment strategies and fully utilize available resources. For example, the BSG [26] guidelines recommend that treatment options for patients who failed to respond to the initial anti-TNF therapy (increase the dose, shorten the dosage interval, switch to an alternative anti-TNF drug or switch to a different drug class) may be informed by the clinical context and by measurement of serum drug and anti-drug antibody concentrations. Nonetheless, this recommendation is weak and based on low-quality evidence. None of the available IBD treatment guidelines provide insights into failure definitions and when to abandon treatment.

It is important to mention that application of clinical trial results to clinical practice is often not straightforward. Issues such as restrictive enrollment criteria, design limitations or conflicts of interest can all underlie the disparity between the outcomes achieved in clinical trials versus those achieved in clinical practice. According to our analysis, drug characteristics (time to clinical response or remission, dosing interval) can make it difficult to implement a universal definition of loss/lack of response and to establish clear timepoints for evaluation. Furthermore, as treatment targets and monitoring tools are constantly improving, we cannot rely on a consensus between society guidelines for decision making. In UC patients, the Mayo score is a widely used and recognized activity index. However, it may not be optimal as some of its components (endoscopic subscore, the Physician Global Assessment) are subjective and introduce variability and a lack of precision into the index. For CD, clinical and endoscopic activity scores are difficult to perform, so they are mostly used in clinical trials. Given the transmural nature of the disease, an accurate evaluation of disease activity would also include cross-sectional imaging techniques, such as computed tomography enterography (CTE) and magnetic resonance enterography (MRE) [28].

In order to formulate universal definitions for loss/lack of response, experts should first reach a consensus regarding clinical and endoscopic response/remission criteria. STRIDE-II, a landmark consensus for IBD management, includes definitions for all proposed treatment targets [29]. Clinical response and clinical remissions definitions are based on patient reported outcomes for both UC and CD and are expressed as percentage changes. An important aspect is that time to achieving treatment targets vary based on therapy and mechanism of action. For example, for anti-TNF the following number of weeks should be allowed between the onset of treatment and assessing proposed targets: 2–4 weeks for clinical response, 4–6 weeks to clinical remission, and 17 weeks for endoscopic healing. On the other hand, for vedolizumab, clinicians should wait 11, 17, and 24 weeks before assessing clinical response, clinical remission, and endoscopic healing, respectively.

At the present time, clinicians abandon treatment after gathering enough data to support their decision (blood and faecal tests, clinical and endoscopic assessments). It would be interesting to investigate in further multicentric prospective studies the strategies that are being used in real life in the therapeutic management of IBD patients. Current monitoring instruments (scores, patient reported outcome measures) were developed as research tools and are too cumbersome for use in daily clinical practice. Another knowledge gap which needs to be filled refers to measuring tools designated for clinical practice and everyday use. Further studies should focus on developing and validating such scores/questionnaires.

### 4.2. Treatment Optimization for Advanced Therapies

Assessment of response to a certain medication is mandatory before declaring treatment failure and moving to another line of treatment. An accurate evaluation can be made by measuring serum drug and ADA concentrations, guided by clinical response and inflammatory markers (blood and faecal) [26]. Measuring serum drug and ADA concentrations helps explain the absence of response that can be attributed to either pharmacokinetic issues, characterized by low drug concentrations with or without the development ADA, or a mechanistic failure in patients with adequate drug concentrations [30]. Several situations can occur:Primary nonresponse: lack of improvement in clinical signs or symptoms after the induction phase. Current guidelines recommend evaluation of response 2–4 weeks after completing loading doses of anti-TNF therapy [26]. Secondary nonresponse or LOR to anti-TNF agents is defined in those patients who initially respond to therapy and subsequently lose the clinical response [31]. In patients that experience LOR to maintenance therapy, TDM should be performed. Current guidelines discuss performing TDM guided by clinical status (reactive approach) or periodically/ at least once during the maintenance period (proactive approach) [28,32]. Other authors defined LOR as patients requiring dose intensification or drug discontinuation after a period of use [33].

Most available data on LOR concern anti-TNF molecules, and newer therapies are not included in guidelines. For example, no studies have provided data thus far on LOR to USTEK in patients with UC. In patients with CD, Yang et al. [34] showed in a recent meta-analysis that primary responders experienced LOR to ustekinumab at a risk of 21% per person-year and required dose escalation at a risk of 25% per person-year. Based on the current LOR definitions, we summarized the current guidelines and recommendations in Table 3, highlighting the discrepancy in management approaches. An ongoing debate exists regarding the usefulness of therapeutic drug monitoring compared with standard therapy, even for well-established therapies such as IFX [35,36]. High-quality studies showed that increasing the dose of infliximab based on a combination of symptoms, biomarkers, and serum drug concentrations does not lead to corticosteroid-free clinical remission in a larger proportion of patients than increasing the dose based on symptoms alone [37]. A summary of current guidelines and recommendations for treatment optimization is listed in Table 3. At present, no formal recommendations are available for the optimization of newer therapies, and confusing data persist regarding anti-TNF drugs (trough level threshold and undefined terms).

### 4.3. Declaring the Failure of Medical Therapy

The decision to stop an advanced therapy is based on clinical and/or endoscopic data after adequate optimization of treatment. As therapeutic options continue to increase and in the absence of clear criteria for a nonresponse and irrecoverable loss of response, therapies might be prematurely abandoned. In contrast, persistent treatment with medical therapies in order to evade surgery may miss the optimal chance of surgical treatment. ECCO guidelines on surgical treatment for acute severe UC recommend colectomy to avoid further increases in surgical morbidity and potential mortality for patients who display no improvement with second-line therapy [38]. The latest ECCO guidelines on CD management recommend always considering surgery as an option for refractory patients [39].

In nonurgent settings, researchers have not clearly determined the number of failed treatment lines that should prompt a recommendation for surgery.

### 4.4. Combining Biologics Strategy

Developing new drugs with different targeted molecular pathways created the premise of combining biologics to maximize efficacy. Available data on the concomitant use of dual biologics are limited to case reports and series of cases. Preliminary evidence suggests that the efficacy of dual biologic therapy might be promising in patients with refractory CD. Vedolizumab and ustekinumab are frequently paired due to their favorable safety profiles. According to a recent study, dual therapy is associated with clinical, biomarker, and endoscopic improvements in patients with refractory CD [40]. A retrospective analysis of 15 patients using either VEDO + IFX, USTE + IFX, or VEDO + USTE identified frequent infections requiring antibiotics, the need for surgical intervention, and hospitalization in a 24-month follow-up period [41]. Results of a systematic analysis totaling 18 patients showed more promising results, with no adverse reactions reported and great clinical outcomes (100% clinical improvement, 93% endoscopic improvement) [42]. Although combination biologic therapy is an attractive strategy, the lack of data mandates future studies prior to its wide use.

### 4.5. Study Limitations

The most important limitation of our study is that data was gathered exclusively from clinical trials methodology and results. Definitions and criteria used in clinical trials are rigorous and apply to select patient populations (previously treated with biologics or refractory to corticosteroids, considered difficult to treat). In common practice, the decision to evaluate and declare treatment failure is tailored to each patient and guidelines do not offer clear recommendations on this subject. Another consequence of the study selection process is that patients with intestinal resections, stomas, or stricturing phenotype were not included in the analysis.

Even though our search gathered data from studies investigating all advanced therapies, some data may have been missed as we relied exclusively on the PubMed (MEDLINE) database for identification of potentially eligible studies.

## 5. Conclusions

Advances in biologic and small-molecule therapeutics have resulted in an increased temptation to prematurely declare failure and switch treatment. The lack of clear formal recommendations on the assessment of a lack/loss of response makes clinical practice heterogeneous and empirical. Debate is ongoing regarding strategies for treatment optimization, and more studies are necessary to establish common practices.

## Figures and Tables

**Figure 1 biomedicines-11-00544-f001:**
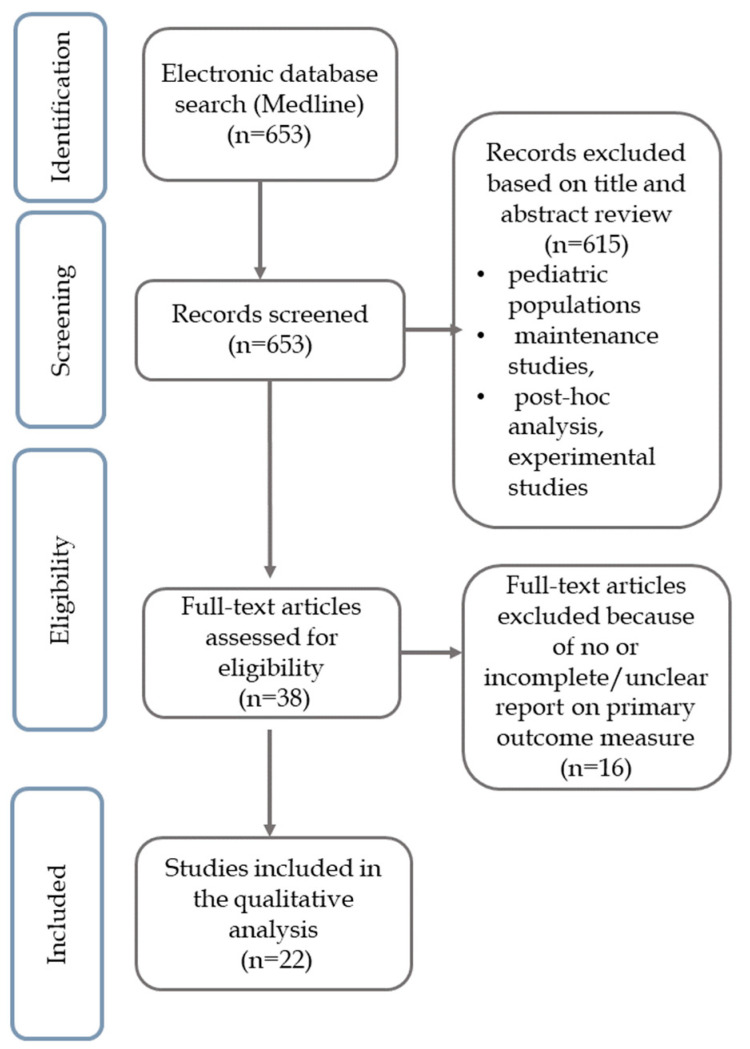
Study selection flowchart.

**Table 1 biomedicines-11-00544-t001:** Definitions of endpoints in ulcerative colitis clinical trials.

Trial/Author	Molecule	Clinical Response	Clinical Remission	Endoscopic Response/ Remission	Time Frame (Weeks)
UNIFI [2]	USTE	decrease in the total Mayo score ≥30% and ≥3 points from baseline, with an accompanying decrease of ≥1 point on the rectal bleeding component of the Mayo scale or a rectal bleeding subscore of 0 or 1	total Mayo score ≤ 2 and no subscore > 1)	Mayo endoscopic subscore ≤1	8
OCTAVE [3]	TOFA	decrease in the total Mayo score ≥30% and ≥3 points from baseline, with an accompanying decrease of ≥1 point on the rectal bleeding component of the Mayo scale or a rectal bleeding subscore of 0 or 1	total Mayo score ≤ 2, with no subscore > 1 and a rectal bleeding subscore of 0	Mayo endoscopic subscore ≤1	8
ACT [4]	IFX	decrease in the total Mayo score ≥30% and ≥3 points from baseline, with an accompanying decrease of ≥1 point on the rectal bleeding component of the Mayo scale or a rectal bleeding subscore of 0 or 1	total Mayo score of 2 points or lower, with no individual subscore exceeding 1 point	Mayo endoscopic subscore ≤1	8
ULTRA [5]	ADM	decrease in the total Mayo score ≥ 30% and ≥3 points from baseline, with an accompanying decrease of ≥1 point on the rectal bleeding component of the Mayo scale or a rectal bleeding subscore of 0 or 1	total Mayo score ≤ 2 points, with no individual subscore exceeding 1 point	Mayo endoscopic subscore ≤1	8
PURSUIT-SC [6]	GOL	decrease in the Mayo score ≥ 30% and ≥3 points from baseline, accompanied by either a rectal bleeding subscore of 0 or 1 or a decrease from baseline in the rectal bleeding subscore ≥1		Mayo endoscopic subscore ≤1	6
UC-SUCCESS [7]	IFX vs. AZA vs. IFX + AZA	decrease in the total Mayo score ≥ 3 points and at least a 30% decrease from baseline	total Mayo score ≤ 2, with no individual subscore exceeding 1 point, without the use of corticosteroids	Mayo endoscopic subscore ≤1	16
NCT00385736 [8]	ADM	decrease in Mayo score ≥ 3 points and ≥ 30% from baseline, plus either a decrease in the rectal bleeding subscore ≥ 1 point or an absolute rectal bleeding subscore of 0 or 1	total Mayo score ≤ 2 and no individual subscore > 1	Mayo endoscopic subscore ≤1	8
SERENE-UC [9]	ADM	decrease in the total Mayo score ≥ 30% and of ≥3 points from baseline, with an accompanying decrease of ≥1 point on the rectal bleeding component of the Mayo scale or a rectal bleeding subscore of 0 or 1	full Mayo score ≤ 2 with no subscore > 1	Mayo endoscopic subscore ≤1	8
Suzuki Y. et al. [10]	ADM	decrease of ≥3 points and ≥30% from baseline plus a decrease in the rectal bleeding subscore [RBS] ≥ 1 or an absolute RBS ≤ 1	full Mayo score ≤2 with no subscore >1	Mayo endoscopic subscore ≤1	8
NCT00787202 [11]	TOFA	decrease in the total Mayo score ≥ 30% and ≥ 3 points from baseline, with an accompanying decrease of ≥1 point on the rectal bleeding component of the Mayo scale or a rectal bleeding subscore of 0 or 1	total Mayo score of 2 points or lower, with no individual subscore exceeding 1 point	Mayo endoscopic subscore of 0	
NCT02039505 [12]	VEDO	reduction of ≥3 points and ≥30% from baseline in the full Mayo score, and a ≥1 point decrease on the rectal bleeding subscore or an absolute rectal bleeding subscore ≤ 1	total Mayo score ≤ 2 and no individual subscore > 1	Mayo endoscopic subscore ≤1	

AZA = azathioprine, IFX = infliximab; ADM = adalimumab; GOL = golimumab; CTZ = certolizumab; VEDO = vedolizumab; USTE = ustekinumab; TOFA = tofacitinib; NA = not applicable.

**Table 2 biomedicines-11-00544-t002:** Definitions of endpoints in Crohn’s disease clinical trials.

Trial/Author	Molecule	Clinical Response	Clinical Remission	Endoscopic Response/Remission	Time Frame (Weeks)
UNITI-2 [13]	USTE	decrease in the CDAI score of at least 100 points from the baseline or a total CDAI score < 150 points (w 6)	CDAI score < 150	decrease in the SES-CD compared with the placebo	8
DIAMOND [14]	ADM vs. ADM + AZA	greater than 70-point reduction in the CDAI score from the baseline value	CDAI score < 150	decrease in the SES-CD of at least 8 points from the baseline, or SES-CD ≤ 4	26
ENTERPRISE [15] (fistulizing CD)	VEDO	≥50% decrease in the number of draining perianal fistulae from the baseline	CDAI score <150	NA	30
NCT00105300 [16]	ADM	decrease in the CDAI score of 70 points or more (70-point response) or of 100 points or more (100-point response) at week 4 compared with the baseline		NA	4
EXTEND [17]	ADM	NA	NA	Absence of ulceration	12
GEMINI-3 [18]	VEDO	decrease in the CDAI score of ≥100 points from the baseline	CDAI score < 150	NA	6
NCT02038920 [19]	VEDO	reduction in the CDAI score ≥ 100 points from the baseline	NA	NA	10
Lemann M. et al. [20]	IFX + AZA	NA	CDAI score < 150	Improvement in the CDEIS	24
NCT00445432 NCT00445939 [21]	IFX	decrease in the CDAI score ≥ 100 or ≥ 70 from the baseline	CDAI score < 150	NA	2 and 4
Sprakes MB et al. [22]	ADM	decrease in the HBI score of 2 points from the baseline value	HBI ≤ 4	NA	6
NCT00615199 [23]	TOFA	decrease in the CDAI score ≥ 70 points from the baseline value	CDAI score < 150	NA	4

AZA = azathioprine, CDEIS = Crohn’s Disease Endoscopic Index of Severity, HBI = Harvey–Bradshaw index; IFX = infliximab; ADM = adalimumab; GOL = golimumab; CTZ = certolizumab; VEDO = vedolizumab; USTE = ustekinumab; NA = not applicable.

**Table 3 biomedicines-11-00544-t003:** Summary of guideline recommendations regarding treatment optimization.

Guideline	Molecule	Trough Level Threshold	ADA	Recommendation
ECCO	IFX/ADM	NA	NA	Insufficient data to make a recommendation on proactive/reactive TDM (CD and UC)
French national consensus [24]	IFX/ADM	<10 μg/mL (12 μg/mL in cases of fistulizing perianal lesions)	absent	Optimize anti-TNF (CD and UC)
		<10 μg/mL (12 μg/mL in cases of fistulizing perianal lesions)	present	Combination therapy with a second anti-TNF agent (CD and UC)
		≥10 μg/mL (12 μg/mL in cases of fistulizing perianal lesions)	absent	Switch to USTE (CD) and VEDO (UC)
AJG Expert Consensus [25]	IFX/ADM/ GOL/CTZ	Depends on the molecule/desired outcomes At least 10–15 μg/Ml for IFX/ADM	High-titer *	Switch within class or out-of-class
			Low-titer	Optimization (dose escalation, interval shortening and/or addition of an immunomodulator)
	VEDO	Depends on the desired outcome and TDM timepoint. No formal recommendation.	Less common (1–4.1%)Titration not recommended	
	USTE	Depends on the desired outcome and TDM timepoint. No formal recommendation.	Less common (0.7–4.6%) Titration not recommended	
BSG [26]	Anti-TNF	optimal **	-	Switch out of class
		Suboptimal **	low	Intensify anti-TNF treatment and consider adding/optimizing an immunomodulator
			intermediate	Intensify anti-TNF treatment and add/optimize immunomodulator
			high	Switch in/out of class. If another anti-TNF agent is used, add/optimize an immunomodulator
	USTE/VEDO/TOFA	NA	NA	NA

IFX = infliximab; ADM = adalimumab; GOL = golimumab; CTZ = certolizumab; VEDO = vedolizumab; TNF = tumor necrosis factor; USTE = ustekinumab; ADA = antidrug antibodies; TDM = therapeutic drug monitoring. * Data are insufficient to support clinically relevant cutoffs to define high-titer antibodies, except a homogeneous mobility shift assay (<10 U/mL). ** Not defined, depends on the assay used and clinical context.

## Data Availability

Not applicable.

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
