# Peer review of "Defining the Failure of Medical Therapy for Inflammatory Bowel Disease in the Era of Advanced Therapies: A Systematic Review"

_biomedicines, 2023, doi:10.3390/biomedicines11020544_

Round 1
Reviewer 1 Report
- - Use oxford comma
- - PROSPERO ID 395550 in not present among protocols in evaluation
- - “A retrospective analysis of 15 patients using either VEDO+IFX, USTE+IFX or VEDO+USTE identified frequent infections requiring antibiotics, the need for surgical intervention and hospitalization in a 24-month follow-up period.”
Other data a more encouraging (cite: “Dual biological therapy with anti-TNF, vedolizumab or ustekinumab in inflammatory bowel disease: a systematic review with pool analysis. Scand J Gastroenterol. 2019 Apr;54(4):407-413. doi: 10.1080/00365521.2019.1597159. Epub 2019 Apr 4. PMID: 30945576.”)
- - Add the limitations of your paper
- - Try to propose a definition of assessment of a lack/loss of response that could be used in all RCT and clinical practice
Author Response
Dear reviewer,
On behalf of all the authors, I would like to thank you for you comments and suggestions. We addressed all the issues raised as follows:
- We added the registration record for PROSPERO. Our protocol is not yet registered, but is being assessed by the editorial team (attached below)
- The citation you recommended was included in the combination therapy section
- We added a paragraph for limitations
- We proposed a definition of assessment of loss of response
The authors,
Reviewer 2 Report
In the present systematic review State et al summarized the main definitions of therapy failure for monoclonal antibodies in IBD, as well as definition of clinical/endoscopic remission and strategies for therapy optimization, showing a relevant heterogeneity.
This is an excellent review which clearly shows definitions used in clinical practice and trials. However, before starting therapy optimization, it would be better to discriminate between primary/secondary non response or loss of response. Therefore definitions of such situations should be added in the text (the definitions in page 9 lines 156-159 are quite vague, and a timing should be reported).
As shown in tables 1-2, heterogeneity exists only for CD. Please underline this.
Author Response
Dear reviewer,
On behalf of all the authors, I would like to thank you for you comments and suggestions. We addressed all the issues raised as follows:
- We expanded the definitions of LOR and added timing considerations recommened in current guidelines
- We underlined the limited heterogencity to CD of LOR definition
The authors,